# Earnings among people with multiple sclerosis compared to references, in total and by educational level and type of occupation: a population-based cohort study at different points in time

Michael Wiberg,[1,2] Chantelle Murley,[1] Petter Tinghög,[1,3] Kristina Alexanderson,[1] Edward Palmer,[4] Jan Hillert,[5] Magnus Stenbeck,[1] Emilie Friberg[1]

For numbered affiliations see end of article.

**Correspondence to**
Chantelle Murley;
chantelle.murley@ki.se

## ABSTRACT

**Objectives** To investigate earnings among people with multiple sclerosis (PwMS) before and after MS diagnosis compared with people without MS, and if identified differences were associated with educational levels and types of occupations. Furthermore, to assess the proportions on sickness absence (SA) and disability pension (DP) in both groups.

**Design** Population-based longitudinal cohort study, 10 years before until 5 years after MS diagnosis.

**Setting** Working-age population using microdata linked from nationwide Swedish registers.

**Participants** Residents in Sweden in 2004 aged 30–54 years with MS diagnosed in 2003–2006 (n=2553), and references without MS (n=7584) randomly selected by stratified matching.

**Outcome measures** Quartiles of earnings were calculated for each study year prior to and following the MS diagnosis. Mean earnings, by educational level and type of occupation, before and after diagnosis were compared using t-tests. Tobit regressions investigated the associations of earnings with individual characteristics. The proportions on SA and/or DP, by educational level and type of occupation, for the diagnosis year and 5 years later were compared.

**Results** Differences in earnings between PwMS and references were observed beginning 1 year before diagnosis, and increased thereafter. PwMS had lower mean earnings for the diagnosis year (difference=SEK 28 000, p<0.05), and 5 years after diagnosis, this difference had more than doubled (p<0.05). These differences remained after including educational level and type of occupation. Overall, the earnings of PwMS with university education and/or more qualified occupations were most like their reference peers. The proportions on SA and DP were higher among PwMS than the references.

**Conclusions** The results suggest that the PwMS' earnings are lower than the references' beginning shortly before MS diagnosis, with this gap increasing thereafter. Besides SA and DP, the results indicate that educational level and type of occupation are influential determinants of the large heterogeneity of PwMS' earnings.

### Strengths and limitations of this study

► The main strengths of this study include the population-based longitudinal cohort design, use of high-quality nationwide registers and a large cohort of people with newly registered multiple sclerosis (MS) diagnoses, which enabled both analyses of earnings on a nationwide scale covering 16 years per individual with MS and comparisons with a matched reference group.

► Potential selection bias was limited by using individual-level information about all residents in Sweden encompassing both the people with MS and the reference group, which is seldom the case for these types of studies.

► Extensive annual data on sociodemographics, occupation and sickness absence and disability pension could be included for 10 consecutive years before the year of MS diagnosis and 5 years after, to further understand the situation before and after MS diagnosis.

► The measure of earnings included the sick pay for employees with short-term sickness absence spells (≤14 days) which might have underestimated the effect of MS on earnings if people with MS have a higher frequency of such spells than the reference group, as lost earnings due to sickness absence are not fully compensated in Sweden.

► Another potential weakness was that we only had information on the MS diagnosis date rather than MS onset date; however, it is unknown whether there are systematic differences among the people with MS regarding the time between onset and diagnosis.

## BACKGROUND

Multiple sclerosis (MS) is a chronic, often progressive, disease that usually onsets in working ages.[1] The combination of the relatively early onset age and chronic nature may affect the individual's economic situation for a substantial part of their working life,

through reduced productivity (and wage) per unit of time worked and/or the number of hours worked.[2–6] Furthermore, MS poses a high cost to society in terms of both direct and indirect costs (from productivity losses).[2 5 7–11] Previous studies have focused predominantly on societal cost burdens, whereas the effects of MS on the development of the individuals' earnings, including possible differences by educational level and type of occupation, have not been extensively researched.

Several studies have found that people with MS (hereafter 'PwMS') are to a greater extent not in paid work compared with people without MS.[3 4 6 12–14] However, sorting individuals into conventional labour market categories (eg, employed/unemployed) obscures more nuanced information about the gradual effect of the disease on the individuals' labour market attachment.

An alternative measure for labour market attachment is the development of individuals' earnings. There are a growing number of studies (including MS-specific[3 6 14–18]) showing negative associations between chronic diseases and individuals' earnings.[3 6 13–25]

The progression of MS is heterogeneous[26 27] and accordingly difficult to predict at the individual level using demographic factors,[28] beyond using age at onset.[29 30] Younger age at onset and educational level have also been shown to influence labour market outcomes among PwMS.[4 31–38] Furthermore, educational level may also suggest the degree of physical labour and/or flexibility at the workplace.[34 36–38]

In working-aged individuals, educational level and type of occupation are highly correlated (eg, university education increases possibilities for more qualified occupations). However, this does not necessarily hold for individuals with chronic diseases, as the disease might hinder them from performing the work corresponding to their educational level. Furthermore, one's educational level may remain unchanged above a certain age whereas an individual's occupation can vary over time, both due to individual and macroeconomic factors. For this reason, in order to analyse the effects of a progressively disabling disease on work, the actual work performed (ie, earnings), may be more informative than the individual's educational level.

Furthermore, PwMS to a high extent work part-time.[2 37 39 40] This is especially relevant for Sweden, where compensation for sickness absence (SA) and disability pension (DP) can be for either part-time or full-time absences from ordinary working hours. Increasing utilisation of these compensation systems (which do not fully compensate the lost earnings) would over time result in increasing differences in annual earnings between PwMS and other wage earners, with all other factors equal.[15] Studies of such associations comparing different time points are still very few, and the possible differences by the individuals' educational level and the type of work remain largely unexplored. The present study aims at improving our knowledge in this respect.

The aims of this study were threefold:

1. To examine levels and distributions of earnings of PwMS compared with population-based matched references, during the years prior to and after the year of diagnosis with MS.
2. To explore whether the level of education and the type of occupation correlate with the within group differences in earnings among PwMS.
3. To compare the proportions of PwMS and matched references covered by part-time or full-time SA and DP benefits.

## METHODS

A population-based longitudinal cohort study was conducted using information from Statistics Sweden on all individuals living in Sweden on 31 December 2004. The study population was attained through two steps.

First, we identified all individuals aged 30–54 years, who in 2003–2006 had their first registered MS diagnosis (ICD-9: 340, ICD-10: G35). Information on MS diagnoses was obtained from the following nationwide registers: inpatient hospital healthcare (data from 1987 to 2006), specialised outpatient healthcare (2001–2006) and outpatient surgery (1997–2000) from the National Patient Register maintained by the National Board of Health and Welfare; and information on SA (2005–2006) and DP (1994–2006) compensated by the Swedish Social Insurance Agency from their Micro-Data for Analysis of the Social Insurance System (MiDAS). Based on the year of their first registered MS code, the PwMS were categorised into four panels: 2003, 2004, 2005 and 2006. Linkages across the registers were conducted by using the unique personal identity number assigned to all residents in Sweden.

Second, a reference group was obtained from the same source population as the PwMS. Group-based stratified matching on the joint distribution of sex and age categories was used to form a comparable reference group for each panel in light of the observed differences in the distribution of PwMS by age and sex. Three references were obtained for each MS patient, and drawn without replacement. The reference individuals had no registered MS diagnosis in 1987–2006.

In total, 2650 PwMS and 7950 reference individuals were included. The year a panel was constituted with newly diagnosed PwMS and reference individuals is, hereafter, referred to as 'the year of inclusion' ($T_0$). Observation spanned from 10 years prior ($T_{-10}$) to the year of inclusion ($T_0$) until 5 years after inclusion ($T_{+5}$).

Sociodemographic and income variables were obtained from the Longitudinal Integration Database for Health Insurance and Labour Market Studies (LISA), maintained by Statistics Sweden, for each of the 16 years studied ($T_{-10}$ until $T_{+5}$) given residence in Sweden at the end of the respective year.

### Outcome variables

Earnings were defined as the annual sum of pre-tax earnings and student allowances. Levels of earnings before

tax were measured over the 16 study years ($T_{-10}$ to $T_{+5}$). Student allowances were included in earnings to limit the risk of underestimating younger individuals' ability to be in paid work in earlier study years. In the Swedish public SA system, the employer usually pays for days 2–14 of an SA spell.[41] This amount could not be disentangled, and was therefore included in our definition of earnings.[42]

All analyses of earnings were performed using 2005 monetary values in Swedish Krona (SEK) inflated by Statistics Sweden's Consumer Price Index. The annual earnings variable was constructed as an untransformed continuous variable.

An additional outcome measure of the annual proportion of individuals on part-time or full-time SA and/or DP was constructed and calculated over the follow-up period. The universal Swedish SA insurance system includes two main benefits: SA and DP. From the age of 16, people with income from work or unemployment benefits can be granted SA if disease or injury leads to reduced work capacity. People aged 19–64 can be granted DP if they have long-term or permanently reduced work capacity due to disease or injury. Both benefits can be granted at four levels (25%, 50%, 75% or 100%) of ordinary working hours, but at different replacement rates (SA: approximately 80%, DP: approximately 64% of lost income, up to a ceiling). Regarding SA, after an initial uncompensated ('waiting') day, the employer pays the next 14 days of the SA spell. A physician's certificate is required by day 8. SA spells are from day 15 covered by the Swedish Social Insurance Agency (from day 2 among unemployed).[41]

### Explanatory variables

The explanatory variables were as follows: sex (female, male), age (categorised in 5-year intervals), country of birth (Sweden, outside of Sweden), educational level (elementary school, high school, university/college) and type of living area (based on population density: smaller municipalities, medium-sized municipalities, larger cities). Further, adopting a definition set by Statistics Sweden (based on annual income and verifications from the workplace) called work status, individuals were categorised as being in work during the year in question (yes, no).[42 43] Values for variables pertained to the 31 December of the year of inclusion, except for the type of living area and work status variables, which were obtained annually.

Annual information on the individuals' type of occupation was also included. Usually, measures of occupation categorise occupations hierarchically, based on the level of specialised skill.[44] Such a categorisation is used by Statistics Sweden with 10 mutually exclusive groups.[42] However, we also aimed to differentiate between types of occupations by how physically demanding they were, and to what extent the individuals were able to have control over their working hours (ie, flexibility at the workplace). Therefore, we collapsed the categories into the following five groups:

1. Manager: Individuals classified as managing a group of workers. Managers performed either office work or manual labour. Therefore, this group was analysed separately.
2. Office work: Non-physical ('white collar') work.
3. Manual labour: Physically demanding ('blue collar') work.
4. Workplace unknown: Registered as in work but with no available information on the workplace.
5. No work: Not categorised as being in work (ie, work status: no).

Individuals identified as employed by the armed forces were excluded from the study population (n=18; <0.1%), due to insufficient information to classify occupation and a very small number of observations.

### Study population

Individuals with missing values for at least one of the sociodemographic variables in the year of inclusion were excluded using list-wise deletion, with the exception of type of occupation. Individuals who had earnings higher than the 99th percentile (in this study) in at least 1 year were also excluded to control for extreme outliers (around 64% of those excluded in this step had earnings above the 99th percentile for more than 1 year). In total, 97 PwMS and 366 references were excluded from the study, due to employment in the armed forces, missing values or having earnings higher than the 99th percentile in a single year. Accordingly, due to these exclusions, the total number of references does not sum to three times the number of PwMS.

The maximum number of years with information for each of the individuals in the study was 16. Due to migration before/after the year of inclusion or death, not all individuals were in the study population for the entire study period. Nonetheless, 94.4% of PwMS and 93.1% of matched references were in the study population for all 16 years. Stratified analyses by country of birth showed that 97.2% of PwMS and 97.5% of the references born in Sweden were included for all 16 years. Corresponding proportions for individuals not born in Sweden were 70.7% and 70.5%, respectively. At the end of follow-up ($T_{+5}$), 2.0% of all PwMS and 2.1% of the references were no longer in the study population (due to migration or death).

### Patient and public involvement

In this explorative observational study, based on population-based de-identified microdata, informed consent was not applicable and PwMS were not involved in the study process. Results are disseminated to PwMS, physicians, and the population through websites and lectures.

### Ethical approval

The project was approved by the Regional Ethical Review Board of Stockholm, Sweden.

## Statistical methods

The levels and distributions of earnings were analysed for PwMS and references, stratified by time before and after the year of inclusion ($T_0$). The four panels were pooled in all analyses.

First, annual mean and median earnings were used to describe the central tendencies among PwMS and references. The annual interquartile range (IQR), the distance between the 25th and 75th percentiles, were used to describe potential converging/diverging trends within, and between, the two groups.

Second, mean earnings before and after $T_0$, stratified by educational level and type of occupation at $T_0$ were calculated. Two-sided t-tests tested the differences in mean earnings between PwMS and the references.

Third, the proportions of individuals on part-time or full-time SA and DP during follow-up, stratified by educational level and type of occupation, were examined for both the PwMS and the references. For individuals with multiple spells of SA or DP in a single year, the spell with the highest grade (25%, 50%, 75% or 100%) was selected.

Finally, Tobit regression models were used to control for potential sociodemographic confounding in analyses and to estimate differences in levels of earnings between PwMS and references. Earnings are a non-negative outcome and often exhibit a skewed distribution with clustering at zero (ie, distinct differences in distributions between those in paid work compared with those not in paid work). Therefore, earnings were used as an observable proxy for individuals' ability to support themselves through their work (ie, earnings above zero). Tobit models avoid the potential bias of analysing outcomes with 'semi-continuous' distributions by ordinary least squares, which may produce negative predicted values due to assumptions of unlimited linearity in the outcome variable.[45] A Tobit model jointly estimates, using a set of explanatory variables, the probability that the individuals' outcome variable is within a predetermined bound and the linear relationship between the explanatory variables and the outcome for individuals within the bound.[46] We used the lower bound zero (ie, no earnings), to observe the actual outcomes, rather than unknown outcomes based on set bound(s), which also means that the observed zeroes were 'genuine zeroes'.[47]

Separate Tobit regressions were estimated at the time points $T_0$ and $T_{+5}$ by two different models. The distributions of earnings for each of these time points were investigated and both exhibited clustering at zero earnings as shown in figure 1. The two models used the same set of explanatory variables (MS (yes/no), panel membership, sex, age at inclusion, country of birth and type of living area), with the exception of level of education and type of occupation, which due to high correlation were analysed separately. Model 1a included the level of education at year of inclusion whereas Model 2a included a time-varying variable for the type of occupation. Furthermore, to estimate the potential additional effects of different levels of education or types of occupation on

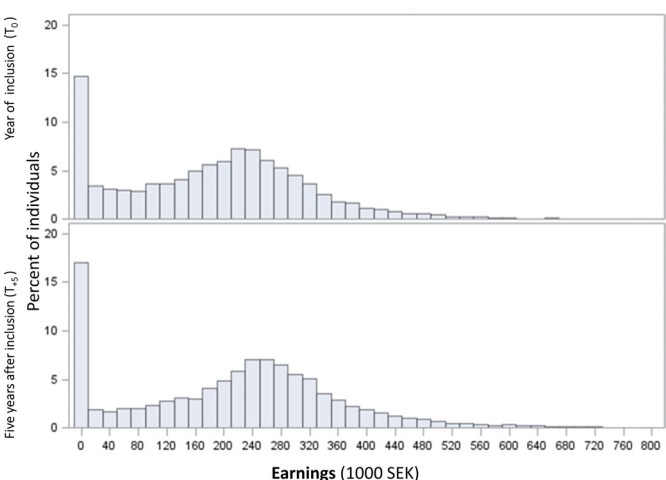

**Figure 1** Distribution of earnings in the total study population, at year of inclusion (n=10 137) and 5 years later (n=9928), in 1000 SEK. Earnings: pre-tax earnings including student allowances, calculated in 2005 monetary values. The year of inclusion ($T_0$) varies between 2003 and 2006. It is the year of the first MS diagnosis for PwMS and the year of inclusion into one of the four panels for references. MS, multiple sclerosis; PwMS, people with MS; SEK, Swedish Krona.

the relationship with levels of earnings, two Tobit models were estimated including two-way interactions between MS and education (Model 1b, university/college as reference) and MS and occupation (Model 2b, Office work as reference), respectively. Statistical tests of whether the estimates deviate enough from zero to reject the assumption of no association were performed at the 95% significance level (p<0.05) and presented as beta estimates with 95% CI. Data management and statistical analyses (regression models) were conducted in SAS V.9.2, with the exception of the differences in mean earnings calculations, which were calculated in Microsoft Excel.

## RESULTS

We identified 2553 PwMS (Panel 2003: 724; 2004: 648; 2005: 636 and 2006: 545) and included 7584 matched references. The distributions of the sociodemographics for the pooled panels at the year of inclusion ($T_0$) are presented in table 1. The majority of the PwMS were women (70%) and were aged 30–45 when diagnosed (62%). In the year of inclusion, the majority had at least some high school education, were born in Sweden (89% among the PwMS compared with 84% in the reference group) and lived in larger cities or medium-sized municipalities. Also, in both groups, most individuals had 'office work' or 'manual labour' as occupations.

There were differences observed between the PwMS and the reference group in the development and intra-year distribution of earnings in the study period (figure 2). The reference group experienced a steady increase in earnings and a predominantly even IQR across the entire period ($T_{-10}$ to $T_{+5}$), whereas the PwMS' trajectory can

**Table 1** Descriptive statistics for the PwMS group and the stratified matched reference group at the year of inclusion $(T_0)$*

| | PwMS | | References | |
| --- | --- | --- | --- | --- |
| | n=2553 | % | n=7584 | % |
| **Sex‡** | | | | |
| Female | 1796 | 70 | 5380 | 71 |
| Male | 757 | 30 | 2204 | 29 |
| **Age‡** | | | | |
| 30–34 | 494 | 19 | 1461 | 19 |
| 35–39 | 531 | 21 | 1576 | 21 |
| 40–44 | 574 | 22 | 1701 | 22 |
| 45–49 | 492 | 19 | 1446 | 19 |
| 50–54 | 462 | 18 | 1400 | 18 |
| **Education level** | | | | |
| University/college | 945 | 37 | 2685 | 35 |
| High school | 1345 | 53 | 3801 | 50 |
| Elementary school | 263 | 10 | 1098 | 14 |
| **Type of occupation** | | | | |
| Managers | 86 | 3 | 259 | 3 |
| Office work | 1082 | 42 | 3020 | 40 |
| Manual labour | 916 | 36 | 2976 | 39 |
| Unknown | 130 | 5 | 352 | 5 |
| No work | 339 | 13 | 977 | 13 |
| **Type of living area** | | | | |
| Larger cities | 960 | 38 | 2789 | 37 |
| Medium-sized municipalities | 902 | 35 | 2681 | 35 |
| Smaller municipalities | 691 | 27 | 2114 | 28 |
| **Country of birth** | | | | |
| Sweden | 2280 | 89 | 6356 | 84 |
| Outside of Sweden | 273 | 11 | 1228 | 16 |
| **Year of inclusion $(T_0)$** | | | | |
| 2003 | 724 | 28 | 2150 | 28 |
| 2004 | 648 | 25 | 1927 | 25 |
| 2005 | 636 | 25 | 1890 | 25 |
| 2006 | 545 | 21 | 1617 | 21 |

*Year of inclusion $(T_0)$ for PwMS: the year of first registered MS diagnosis. Year of inclusion $(T_0)$ for references: the year of inclusion to one of the four panels.
‡ Variables used for stratified matching ($1_{MS}$->$3_{REF}$).
MS, multiple sclerosis; PwMS, people with MS.

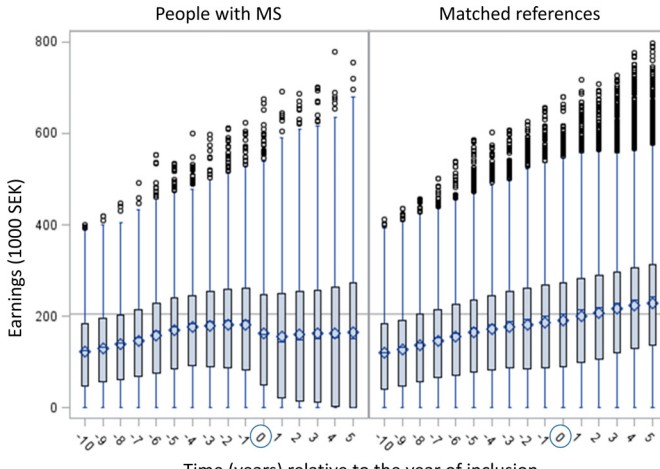

**Figure 2** Distribution (box and whisker plots) of earnings by time, for PwMS (n=2553) and matched references (n=7584), in 1000 SEK. Earnings: pre-tax earnings including student allowances, calculated in 2005 monetary values. The year of inclusion $(T_0)$ varies between 2003 and 2006. It is the year of the first MS diagnosis for PwMS and the year of inclusion into one of the four panels for references. Observations that were larger than 1.5*IQR are circled. Symbols in the box plot: bar=median earnings, diamond=mean earnings. Line across box-plots=median earnings for the reference group at $T_0$ (SEK 204 850). MS, multiple sclerosis; PwMS, people with MS; SEK, Swedish Krona.

years $T_{-1}$, $T_0$ and $T_{+5}$ (table 2). These differences in mean annual earnings increased with time. In $T_0$, PwMS' mean earnings were 85% of the references' mean earnings, whereas the corresponding proportion in $T_{+5}$ was 73%.

Earnings stratified by educational level and differences in earnings between the PwMS and the corresponding reference subgroup were measured for the study years $T_{-5}$, $T_{-2}$, $T_{-1}$, $T_0$ and $T_{+5}$ (table 2). PwMS with at most elementary or high school education, experienced a decrease in mean earnings from $T_{-1}$ and onwards. For PwMS with university/college education, there was a decrease in mean earnings from $T_{-1}$ to $T_0$, but an increase from $T_0$ to $T_{+5}$. The reference group had increasing trends in mean earnings for all levels of education, which resulted in increasing differences in mean earnings between PwMS and the references. In the last study year $(T_{+5})$, the largest differences in earnings were found among those with at most elementary school education, where the PwMS' mean earnings were SEK 84 000, corresponding to 53% of the mean earnings of their peers in the reference group. Furthermore, PwMS with high school and university/college education also had lower earnings than their peers in the reference group. The corresponding differences and proportions for high school and university/college education were SEK 68 000 (68%) and SEK 63 000 (78%), respectively.

Likewise, differences in earnings between the PwMS and the references at $T_{-5}$, $T_{-2}$, $T_{-1}$, $T_0$ and $T_{+5}$ were stratified by type of occupation at inclusion (table 2). In all time periods, for both PwMS and references, the mean

be divided into three periods: $T_{-10}$ to $T_{-2}$; $T_{-1}$ to $T_{+1}$; and $T_{+2}$ to $T_{+5}$. The first period ($T_{-10}$ to $T_{-2}$) was characterised by increasing earnings for PwMS with an even intra-year distribution (measured by IQR). In the second period ($T_{-1}$ to $T_{+1}$), PwMS' earnings were decreasing and the IQR was diverging. In the third period ($T_{+2}$ to $T_{+5}$), the level of earnings was stable but IQR diverging.

When comparing differences in mean levels of earnings among PwMS and references, overall, PwMS had lower earnings compared with the references for the study

**Table 2** Mean earnings* of PwMS and differences in mean earnings between the PwMS group and matched reference group at different lengths in time from the year of inclusion ($T_0$), stratified by educational level and type of occupation at $T_0$

| | $T_{-5}$ | $T_{-2}$ | $T_{-1}$ | $T_0$ | $T_{+5}$ |
|---|---|---|---|---|---|
| | PwMS† (diff)‡ | PwMS† (diff)‡ | PwMS†(diff)‡ | PwMS† (diff)‡ | PwMS† (diff)‡ |
| All | 169 (4) | 182 (1) | 181 (−6)§ | 163 (−28)§ | 166 (−63)§ |
| Educational level | | | | | |
| University/college | 192 (5) | 214 (1) | 214 (−6) | 203 (−24)§ | 218 (−63)§ |
| High school | 159 (0) | 168 (−6) | 166 (−12)§ | 146 (−37)§ | 145 (−68)§ |
| Elementary school | 135 (9) | 142 (12) | 135 (3) | 111 (−25)§ | 84 (−74)§ |
| Type of occupation | | | | | |
| Managers | 251 (−12) | 308 (−5) | 308 (−24) | 306 (−27) | 258 (−113)§ |
| Office work | 207 (2) | 230 (−5) | 233 (−11)§ | 219 (−33)§ | 226 (−65)§ |
| Manual labour | 163 (6) | 179 (4) | 178 (−4) | 152 (−36)§ | 146 (−71)§ |
| Unknown | 96 (6) | 90 (−5) | 99 (−7) | 102 (−18) | 88 (−69)§ |
| No work | 65 (13)§ | 39 (10)§ | 20 (6)§ | 4 (0) | 25 (−27)§ |

*Earnings: pre-tax earnings including student allowances, presented in 2005 values as 1000 SEK.
†PwMS (n=2553); References (n=7584).
‡Differences in mean earnings (SEK 1000) between PwMS and references in parentheses. Calculated with two-sided t-tests with unequal variance and unequal number of observations.
§Differences significantly different from zero (p<0.05).
PwMS, people with multiple sclerosis; SEK, Swedish Krona.

earnings were highest among the managers (table 2). Further, office workers had higher mean earnings compared with those with manual work. These findings reflected those that we obtained for level of education. PwMS in manual work had lower earnings compared with their reference counterparts in the final study year ($T_{+5}$) (67%). The corresponding figure for office workers was 78%. However, the largest difference in earnings in the final study year was observed among managers, with SEK 113 000 less among PwMS than their peers, which corresponded to 70% of the mean earnings of the reference group managers. Finally, those with no work at inclusion had much lower levels of earnings compared with those with known occupations both prior to and after this time point.

The proportions of individuals with at least one spell of part-time or full-time SA and similarly DP, were larger among the PwMS than in the corresponding group among the references for both $T_0$ and $T_{+5}$ (table 3). In the diagnosis year ($T_0$), 46.7% of the PwMS had at least one full-time SA spell and 10.0% had part-time SA. Among PwMS, the proportion of part-time and full-time SA decreased with time, whereas the proportion of DP increased substantially. In the last year of follow-up, 21.0% of the PwMS had part-time DP, and the proportion of full-time DP was 25.1%. The patterns were similar for the reference group, but at lower levels than among the PwMS.

For PwMS, the proportions of part-time and full-time SA at $T_{+5}$ were somewhat similar for all levels of education; however, the proportions varied more by type of occupation (table 3). Among the PwMS, managers and individuals with no work had lower proportions of part-time and

full-time SA than PwMS in the other types of occupations. For PwMS in the last study year, there was a larger proportion of individuals with part-time DP than full-time DP among those with university/college education and with more qualified types of occupations.

The estimates from the regression Models 1a, including educational level, and 2a, including type of occupation, revealed a negative association between MS and earnings (table 4). In $T_0$, the estimates for MS (yes/no) showed similar negative relationships with earnings when comparing the regression that included educational level ($T_0$: −36.9; 95% CI: −43.0,−30.8) and that which included type of occupation ($T_0$: −33.3; 95% CI: −38.0,−28.6). The corresponding estimates for 5 years later ($T_{+5}$) were also both negative, but with a larger difference between the estimates when including educational level ($T_{+5}$: −84.4; 95% CI: −91.5,−77.3) compared with type of occupation ($T_{+5}$: −49.9; 95% CI: −55.6,−44.2).

A hierarchic pattern in the estimates for educational level and type of occupation was observed. Higher levels of education (ie, university/college) and more qualified occupations (ie, managers and office workers) were associated with relatively higher levels of earnings compared with those with lower levels of education (ie, high school and elementary school) or less qualified occupations.

In the additional Tobit regressions for earnings in the year of inclusion and 5 years later (Models 1b and 2b), interactions between MS and education or occupation, respectively, were included (online supplementary table 1). In Model 1b, we observed that the interaction between high school education and MS was negative in $T_0$ (−16.9; 95% CI: −29.6,−4.2). In $T_{+5}$, the interaction between

**Table 3** Percentages* of the PwMS and matched reference group who had SA† and DP† in the year of inclusion (T₀) and 5 years later (T₊₅), stratified by educational level and type of occupation at T₀

| Year | Factor | Category | Sickness absence | | | | Disability pension | | | |
| | | | PwMS‡ | | Ref‡ | | PwMS‡ | | Ref‡ | |
| | | | Part-time | Full-time | Part-time | Full-time | Part-time | Full-time | Part-time | Full-time |
|---|---|---|---|---|---|---|---|---|---|---|
| T₀ | Education | University/college | 11.0 | 43.0 | 2.6 | 10.0 | 3.6 | 3.8 | 1.5 | 2.0 |
| | | High school | 9.1 | 49.5 | 2.1 | 15.6 | 5.7 | 10.0 | 2.5 | 6.4 |
| | | Elementary school | 11.0 | 45.6 | 1.3 | 16.3 | 8.4 | 22.8 | 3.5 | 16.9 |
| T₊₅ | | University/college | 7.2 | 17.7 | 1.6 | 8.4 | 21.6 | 15.4 | 2.1 | 2.8 |
| | | High school | 5.3 | 17.1 | 0.8 | 11.9 | 20.9 | 27.3 | 3.0 | 8.1 |
| | | Elementary school | 6.6 | 13.2 | 0.8 | 10.0 | 19.7 | 48.3 | 4.9 | 19.6 |
| T₀ | Occupation | Managers | 16.3 | 31.4 | 0.8 | 5.8 | 1.2 | 0.0 | 1.2 | 0.4 |
| | | Office work | 13.3 | 43.6 | 2.9 | 11.0 | 5.4 | 1.9 | 2.0 | 0.7 |
| | | Manual labour | 8.5 | 55.7 | 2.1 | 17.2 | 5.1 | 3.0 | 2.7 | 1.4 |
| | | Unknown | 10.0 | 43.9 | 2.3 | 17.1 | 12.3 | 13.1 | 2.3 | 7.7 |
| | | No work§ | 2.1 | 37.2 | 0.3 | 15.0 | 3.2 | 49.3 | 2.4 | 39.8 |
| T₊₅ | | Managers | 4.7 | 8.2 | 0.4 | 4.7 | 24.7 | 15.3 | 1.2 | 0.8 |
| | | Office work | 8.0 | 18.7 | 1.5 | 8.7 | 25.5 | 12.7 | 2.5 | 1.5 |
| | | Manual labour | 5.4 | 19.1 | 0.8 | 13.6 | 20.2 | 23.7 | 3.5 | 2.9 |
| | | Unknown | 7.9 | 13.4 | 1.7 | 11.3 | 24.2 | 33.6 | 2.9 | 10.4 |
| | | No work§ | 1.9 | 8.5 | 0.8 | 6.8 | 6.3 | 69.7 | 3.2 | 45.5 |
| T₀ | All | – | 10.0 | 46.7 | 2.1 | 13.7 | 5.2 | 9.1 | 2.3 | 6.3 |
| T₊₅ | | – | 6.2 | 16.9 | 1.1 | 10.4 | 21.0 | 25.1 | 2.9 | 7.9 |

*Row percent of all individuals (including those without SA or DP in each subgroup).

†Individuals with a spell of SA or DP with the highest grade (25%, 50%, 75% or 100%) at T₀ and T₊₅ (ie, if an individual had both part-time and full-time SA in a single year, the individual was categorised as full-time SA; the same procedure was used for individuals on DP).

‡Numbers at year T₀: PwMS (n=2553) and references (n=7584).

§Individuals with unemployment benefits can be granted SA if the disease/injury is severe enough to prevent them from seeking employment. DP can be granted until the age of 65. In most cases, SA refers to compensation for an SA spell >14 days (the first 14 are covered by the employer).

DP, disability pension; PwMS, people with multiple sclerosis; SA, sickness absence.

**Table 4** Cross-sectional Tobit regressions for earnings* for the year of inclusion† and 5years later

| | Education | | Occupation | |
| --- | --- | --- | --- | --- |
| | Model 1a | | Model 2a | |
| | Year of inclusion (T$_0$) Estimated difference‡ | Five years later (T$_{+5}$) Estimated difference‡ | Year of inclusion (T$_0$) Estimated difference‡ | Five years later (T$_{+5}$) Estimated difference‡ |
| | (95% CI) | (95% CI) | (95% CI) | (95% CI) |
| Intercept | 217.3 (208.3 to 226.3)§ | 285.8 (275.2 to 296.4)§ | 241.7 (234.6 to 248.8)§ | 279.4 (271.2 to 287.6)§ |
| MS (ref: No MS)¶ | −36.9 (−43.0 to −30.8)§ | −84.4 (−91.5 to −77.3)§ | −33.3 (−38.0 to −28.6)§ | −49.9 (−55.6 to −44.2)§ |
| Male (ref: Female)¶ | 68.1 (62.4 to 73.8)§ | 67.8 (61.1 to 74.5)§ | 63.4 (58.9 to 67.9)§ | 64.2 (58.9 to 69.5)§ |
| Age (ref: 30–34years)¶ | | | | |
| 35–39 | 17.5 (9.3 to 25.7)§ | 20.6 (11.0 to 30.2)§ | 9.6 (3.1 to 16.1)§ | 18.2 (10.8 to 25.6)§ |
| 40–44 | 35.5 (27.5 to 43.5)§ | 25.1 (15.7 to 34.5)§ | 25.7 (19.2 to 32.2)§ | 27.4 (20.0 to 34.8)§ |
| 45–49 | 33.4 (25.0 to 41.8)§ | 6.1 (−3.9 to 16.1) | 24.3 (17.6 to 31.0)§ | 16.6 (8.8 to 24.4)§ |
| 50–54 | 30.0 (21.6 to 38.4)§ | −9.4 (−19.4 to 0.6) | 22.3 (15.6 to 29.0)§ | 9.6 (1.6 to 17.6)§ |
| Country of birth (ref: Sweden)¶ | −78.4 (−86.0 to −70.8)§ | −71.7 (−80.7 to −62.7)§ | −20.3 (−26.6 to −14.0)§ | −19.9 (−27.3 to −12.5)§ |
| Panel (ref: 2006)¶ | | | | |
| 2003 | −5.1 (−12.5 to 2.3) | −9.4 (−18.2 to −0.6)§ | −6.9 (−12.8 to −1.0)§ | −9.3 (−16.2 to −2.4)§ |
| 2004 | −5.7 (−13.3 to 1.9) | −2.1 (−11.1 to 6.9) | −7.3 (−13.4 to −1.2)§ | −2.5 (−9.6 to 4.6) |
| 2005 | −7.8 (−15.4 to −0.2)§ | −6.7 (−15.7 to 2.3) | −6.0 (−12.1 to 0.1) | −3.1 (−10.2 to 4.0) |
| Type of living area (ref: Larger cities)** | | | | |
| Medium-sized municipalities | −19.9 (−26.0 to −13.8)§ | −23.3 (−30.6 to −16.0)§ | −20.8 (−25.7 to −15.9)§ | −21.9 (−27.6 to −16.2)§ |
| Smaller municipalities | −26.1 (−32.8 to −19.4)§ | −32.7 (−40.5 to −24.9)§ | −29.5 (−34.8 to −24.2)§ | −31.6 (−37.9 to −25.3)§ |
| Educational level (ref: University/college)¶ | | | | |
| High school | −58.1 (−63.8 to −52.4)§ | −79.7 (−86.4 to −73.0)§ | - | - |
| Elementary school | −113.2 (−121.8 to −104.6)§ | −148.6 (−158.8 to −138.4)§ | - | - |
| Type of occupation (ref: Office work)** | | | | |
| Managers | - | - | 63.5 (52.3 to 74.7)§ | 87.8 (76.8 to 98.8)§ |
| Manual labour | - | - | −65.6 (−70.1 to −61.1)§ | −72.5 (−77.6 to −67.4)§ |
| Unknown | - | - | −135.9 (−145.5 to −126.3)§ | −171.6 (−187.3 to −155.9)§ |
| No Work | - | - | −384.3 (−394.9 to −373.7)§ | −519.7 (−538.9 to −500.5)§ |
| Sigma | 131.2 (129.6 to 132.9)§ | 152.4 (150.4 to 154.4)§ | 100.7 (99.4 to 102.0)§ | 113.8 (112.3 to 115.3)§ |
| N | 10137 | 9928 | 10137 | 9928 |

Continued

**Table 4** Continued

| | Education Model 1a | | Occupation Model 2a | |
|---|---|---|---|---|
| | Year of inclusion ($T_0$) Estimated difference‡ (95% CI) | Five years later ($T_{+5}$) Estimated difference‡ (95% CI) | Year of inclusion ($T_0$) Estimated difference‡ (95% CI) | Five years later ($T_{+5}$) Estimated difference‡ (95% CI) |
| N censored at zero | 1260 | 1505 | 1260 | 1505 |
| BIC | 114 727 | 111 972 | 107 869 | 104 307 |

*Earnings: pre-tax earnings including student allowances. Presented in 2005 values, in SEK 1000.
†Year of inclusion ($T_0$) for PwMS: the year of first MS diagnosis. Year of inclusion ($T_0$) for references: the year of inclusion to one of four panels. Estimates from Tobit models with lower bound set at zero (ie, no earnings).
‡Beta estimate.
§Differences significantly different from zero (p<0.05).
¶Time-invariant variables: measured at $T_0$.
**Time-variant variables: measured at $T_0$ or $T_{+5}$.
BIC, Bayesian information criterion; MS, multiple sclerosis; PwMS, people with MS; SEK, Swedish Krona.

elementary school education and MS was also negative (−42.7; 95% CI: −68.0,−17.4). Further, interactions between occupation and MS (Model 2b) found a negative association among managers in $T_{+5}$ (-40.7; 95% CI: −68.5,−12.9). Whereas a non-significant negative association was found among manual labour in both $T_0$ (−5.0; 95% CI: −15.2, 5.2) and $T_{+5}$ (−5.6; 95% CI: −17.8, 6.6).

## DISCUSSION

In this cohort study, we identified 2553 PwMS of working ages and compared them with 7584 references without MS across time points for a period of 16 years. The principal findings were that both groups had similar levels of earnings and annual changes in earnings up until 1 year before the diagnosis year; thereafter, the growth in earnings levelled off among the PwMS for the remainder of the study period, while earnings for the references continued to increase. These differences remained when we controlled for a number of sociodemographic variables. The largest differences in absolute terms between PwMS and the references were observed for those with elementary school education and for those working as managers. Those with elementary school education also had the largest relative differences. There were also large within-group differences in the development of mean earnings for the PwMS with respect to educational level and type of occupation. Additionally, there were higher proportions among PwMS of part-time and full-time SA and DP after the year of MS diagnosis compared with among the references.

This study has several strengths, including identifying the PwMS from nationwide patient registers.[48] Furthermore, the data from the nationwide registers are of high quality,[49] which made it possible to conduct our analyses with no loss to follow-up and without self-selection bias. One limitation is that the onset of MS always precedes the diagnosis, and this time lag varies considerably depending on the experienced symptoms, the individual's healthcare-seeking behaviour, healthcare practices and timeliness to confirm diagnosis.[2 50] Therefore, the year of diagnosis used in this study can be seen as a biased proxy (although with known direction) for the year of MS onset. Without additional MS-specific data on clinical progression, we could not evaluate the degree to which this time lag affected the findings. The extent to which the time lag correlates with sociodemographic factors is, to the best of our knowledge, not known. Furthermore, educational level and type of occupation were highly correlated, and therefore, analysed separately. Hence, when we analysed one of the variables, the effect of the other variable could, to some degree, be incorporated indirectly. The generalisability of the findings may be limited to countries with a similarly functioning labour market and welfare system. Finally, due to data limitations, our definition of earnings includes shorter spells of SA (days 2–14), which may conceal the initial effects of MS, and underestimate the

potential effect of MS on earnings for individuals with repeated short SA spells.

Our overall results are in line with previous studies of PwMS, which have shown non-increasing earnings,[15] and high levels of unemployment or SA/DP after the MS diagnosis.[3 13 14 32 34 38 51–53] We found that differentiating the analyses by either educational level or type of occupation overall gave similar results. Nonetheless, the combination of occupation and annual earnings provided additional information, including information on those who were working compared with those not in paid work. Furthermore, by comparing the PwMS' earnings with that of their reference peers by occupation, we showed that PwMS classified as managers had larger differences in earnings than those in office work. However, it is possible that the large aggregates chosen for categorisation of occupation disguised within-group variation in terms of the physical demands of work tasks. Likewise, the lack of differentiation within the university/college educational category could have concealed strong heterogeneity.

We found considerable heterogeneity within the group of PwMS; while a large proportion appeared to participate on the labour market to a high extent, many had low or no earnings at all. There are many possible explanations for this heterogeneity, with the individuals' educational level and type of occupation as two influential factors within this study. This suggests that the PwMS' work situations are important determinants of subsequent labour market outcomes. We also observed that PwMS with university/college education or with more qualified types of occupations (ie, office workers or managers) had higher proportions of part-time DP (rather than full-time DP) than those with lower education levels or less qualified types of work (ie, manual workers or those with no work). This could potentially reflect the nature of physical work demands among manual occupations or reflect a lower degree of flexibility to adjust one's work situation than among non-manual occupations, which consequently may lead to higher rates of full-time rather than part-time DP. It was not possible to capture this distinction in the present study. However, it is also possible that our variables, educational level and type of occupation, were proxies for less tangible individual factors. One such factor could be healthcare-seeking behaviour, where those with a higher educational level or more qualified occupations may receive the MS diagnosis, and thus treatment, earlier in the disease trajectory. Nevertheless, MS affected an individual's work capacity regardless of whether a diagnosis had been set or not. The new generation of disease-modifying treatments to slow disease progression, and thus preserve work capacity, were first introduced in 2006, but not administered to the majority of PwMS until years later.[54] Therefore, it is possible that this could partly explain the observed differences in earnings between the groups. Similarly, the wide distribution of earnings among PwMS could reflect the known heterogeneous progression of the disease—irrespective of educational level or type of occupation.

Furthermore, the large differences in earnings among individuals with educational levels less than university/college following MS diagnosis compared with the reference group, provides support for the conclusion that the degree of physical labour and/or flexibility at the work place is of importance for remaining in paid work once diagnosed with MS. However, further research is required to explain the observed differences between PwMS and the references in managerial positions.

A possible explanation of the observed increasing differences in earnings between PwMS and individuals without MS, in addition to working fewer hours, and situations of pay for performance, could be due to different opportunities for mobility in the labour market. An individual's wage development is generally associated with possibilities for changing employers. This is especially so in Sweden, with largely centrally negotiated wage increases and often limited individual performance variation.[55 56] Thus, labour market mobility provides an opportunity for substantial wage improvement, and an MS diagnosis may deter changing employment.[57] More studies of PwMS' or more generally, on chronically ill individuals' wages and labour market mobility are needed to increase our knowledge about such mechanisms, for example, by studying transitions between different occupations, types of employment and unemployment.[58 59]

Levels and development of earnings are known to differ by both age and sex.[60] Accordingly, in the present study, we used both of these variables in selecting our matched reference group. Future studies focused on how a diagnosis with MS leads to deviations from the expected earnings trajectory by age and sex are warranted. Trajectories of PwMS' mean earnings stratified by age group and sex in a previous study suggest that there are differences in both patterns and levels of earnings among PwMS.[61] However, more detailed analyses, including a reference group, are needed. This is especially so given that the progression of MS is to some degree age-dependent.[30]

Lastly, health outcomes research is limited by a lack of measures reflecting the multitude of MS consequences.[62] The traditional MS-specific health outcome measurement tool, Expanded Disability Systems Score (EDSS), typically overemphasises the importance of motor function and poorly reflects cognitive functions and fatigue,[62] two factors strongly influencing work capacity.[18 62–64] Our results suggest that earnings could potentially be a promising outcome measure assessing a composite of functions.[18] The underlying assumption is that the reduction of earnings among PwMS to a great extent reflects MS-related disability. The observed impact of MS on earnings among PwMS across occupational types and educational levels indicates strongly that motor function, which is known to deteriorate slowly, at least in younger patients, is not the sole reason for the decrease in earnings, thus, indicating the possible influence of fatigue and cognition.[18 63] Furthermore, our findings of earnings development reflect the same trends as the pattern of disability development, as assessed by EDSS, suggesting

that earnings may serve as a surrogate outcome measure reflecting the broader consequences of MS.[18] Earnings, when available from public registers, such as from tax authorities, have the additional benefits of minimal data loss and reflecting comprehensive time periods, whereas EDSS outcomes reflect single points in time.[18] Further studies are required with data sets containing both earnings and EDSS scores to compare the two in terms of resolution and validity.

## CONCLUSIONS

From 1 year prior to the MS diagnosis and onwards, we observed increasingly higher levels of mean earnings among the reference group compared with among the PwMS. Further, we observed considerable within-group heterogeneity among the PwMS. After 5 years of being diagnosed, a large percentage of PwMS still participated in the labour market, nevertheless, many had low or no earnings at all. We also found that those with at most elementary or high school levels of education or less qualified types of occupations were less alike their respective peers in the reference group than those with university/college levels of education and more qualified types of occupations. Moreover, part-time and full-time SA and DP were more common among the PwMS—a result that could potentially be used to describe the observed heterogeneity of earnings among the PwMS.

**Author affiliations**
¹Division of Insurance Medicine, Department of Clinical Neuroscience, Karolinska Institutet, Stockholm, Sweden
²Department of Analysis and Forecast, Swedish Social Insurance Agency, Stockholm, Sweden
³Department of Health Sciences, Swedish Red Cross University College, Huddinge, Sweden
⁴Uppsala Center for Labor Studies, Department of Economics, Uppsala University, Uppsala, Sweden
⁵Division of Neurology, Department of Neuroscience, Karolinska Institutet, Stockholm, Sweden

**Contributors** MW, PT, KA, EP, MS and EF participated in the design of the study. MW performed the data management and performed all analyses. All authors (MW, CM, PT, KA, EP, JH, MS and EF) contributed to the interpretations of the results. MW and CM drafted the manuscript. PT, KA, EP, JH, MS and EF contributed by revising for important intellectual content. All authors have approved the final version of the manuscript.

**Funding** The study had financial support from unrestricted research grants from Biogen and the Swedish Research Council for Health, Working Life and Welfare.

**Competing interests** MW, CM, PT and EF were partly funded by Biogen. EP and MS have no competing interests to declare. KA has received unrestricted grants from Biogen. JH received honoraria for serving on advisory boards for Biogen and Novartis and speaker's fees from Biogen, Merck-Serono, Bayer-Schering, Teva and Sanofi-Aventis. He has served as PI for projects sponsored by, or received unrestricted research support from, Biogen, Merck-Serono, TEVA, Novartis and Bayer-Schering. His MS research is funded by the Swedish Research Council. The design of the study, data collection, analyses, interpretations of data and writing of manuscript were performed without involvement of the funding bodies. Biogen was given the opportunity to comment on the manuscript before submission.

**Patient consent for publication** Not required.

**Ethics approval** The project was approved by the Regional Ethical Review Board, Stockholm, Sweden.

**Provenance and peer review** Not commissioned; externally peer reviewed.

**Data sharing statement** Not applicable. The data cannot be made publically available due to privacy regulations. According to the General Data Protection Regulation, the Swedish, the Swedish Data Protection Act, the Swedish Ethical Review Act and the Public Access to Information and Secrecy Act, data can be made available only for specific purposes, including research that meets the criteria for access to this type of sensitive and confidential data as determined by a legal review. Readers may contact Professor Kristina Alexanderson (kristina. alexanderson@ki.se) regarding the data.

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
