## [Reviewer comments · BMJ Open]

ARTICLE DETAILS

TITLE (PROVISIONAL)	Earnings among people with multiple sclerosis compared to references, in total and by educational level and type of occupation: a population-based cohort study at different points in time
AUTHORS	Wiberg, Michael; Murley, Chantelle; Tinghög, Petter; Alexanderson, Kristina; Palmer, Edward; Hillert, Jan; Stenbeck, Magnus; Friberg, Emilie

VERSION 1 - REVIEW

REVIEWER	John Pearson University Otago Christchurch New Zealand
REVIEW RETURNED	13-Jul-2018

GENERAL COMMENTS	Very nice dataset that directly quantifies the loss of earnings between people with MS and their peers. Longitudinal data like this is a particularly strong addition to the literature. The data has been appropriately analysed and conclusions are well supported by the data. The authors have struck a good balance between sufficiently describing their statistical methods and maintaining readability, although I have asked below for a little more detail. It is also usual to include what software was used for the analysis. As it stands this is an interesting addition to the literature, however I'll take the opportunity to ask the authors to dig a little deeper. Can you include an anova type analysis for each model to demonstrate statistical support for each factor in the models. Were there any interaction effects, particularly between MS and Age, MS and Education, MS and Occupation? That is, does your data support a difference in the earnings~age curve for MS and non-MS? It is possible and likely given figure 2, that the relationship is non linear so perhaps GEEs or non linear effects might be worth trying. It would also be worth exploring these differential effects for the other covariates. There is some evidence in the literature for sex differences too, and I think your data could be used to address this, for example do females with higher qualifications have a greater loss of income than their male counterparts in this dataset? minor points:
--

	I 50: the estimate is SEK28,000 but the CI limits are negative, this is a bit awkward, I would just make them all positive and note that you did say "lower earnings" to give direction. I 54 alike should be like I 86 "may" I think you can cite literature to make a stronger claim that it does. Table 4 N censored at zero reads 1,26
--	--

REVIEWER	Claudia Pflieger Neurological Department Aalborg University Hospital Denmark
REVIEW RETURNED	09-Dec-2018

GENERAL COMMENTS	Dear authors Thank you for this nicely written and systematically constructed paper on "earnings among people with multiple sclerosis compared to references, in total and by educational level and type of occupation: a population-based cohort study at different points in time" Your findings are much in line with previous research and adds important knowledge to this interesting topic.
---

VERSION 1 – AUTHOR RESPONSE

Reviewer: 1

Reviewer Name: John Pearson

Institution and Country: University Otago Christchurch - New Zealand Please state any competing interests or state 'None declared': None declared

1. Very nice dataset that directly quantifies the loss of earnings between people with MS and their peers. Longitudinal data like this is a particularly strong addition to the literature. The data has been appropriately analysed and conclusions are well supported by the data. The authors have struck a good balance between sufficiently describing their statistical methods and maintaining readability, although I have asked below for a little more detail. It is also usual to include what software was used for the analysis. As it stands this is an interesting addition to the literature, however I'll take the opportunity to ask the authors to dig a little deeper.

Authors' response: Thank you very much.

We have now revised the Methods section (See page 12-13, lines 280-283) to include the software used: SAS v9.2 in the data management and regression models and Excel for the differences in mean earnings calculations presented in Table 2.

Regarding the suggestion to 'dig a little deeper' please see our response to the next comment.

2. Can you include an anova type analysis for each model to demonstrate statistical support for each factor in the models. Were there any interaction effects, particularly between MS and Age, MS and Education, MS and Occupation? That is, does your data support a difference in the earnings~age curve for MS and non-MS? It is possible and likely given figure 2, that the relationship is non linear so perhaps GEEs or non linear effects might be worth trying. It would also be worth exploring these differential effects for the other covariates. There is some evidence in the literature for sex differences too, and I think your data could be used to address this, for example do females with higher qualifications have a greater loss of income than their male counterparts in this dataset?

Authors' Response: Thank you for the suggestion to delve deeper into the data with your engaged response.

We have now included models which consider the interactions between MS*Education (Model 1b) and MS*Occupation (Model 2b) as requested (See page 12, lines 274-278 in Methods, and page 20, lines 389-397 in Results). We have included the table presenting these interactions below for your interest, however, in the interest of brevity we have summarised the key findings in text within the revised manuscript.

Table 1 (Reviewer response): Cross-sectional Tobit regressions for earnings in the year of inclusion and five years later, with interactions.

Education

Model 1b Occupation

Model 2b

T0 T+5 T0 T+5

Beta estimate Beta estimate Beta estimate Beta estimate

(CI 95%) (CI 95%) (CI 95%) (CI 95%)

Intercept 214.7 (205.5, 223.9) 282.8 (271.8, 293.8) 241.8 (234.5, 249.1) 278.2 (270.0, 286.4)

MS (ref: No MS)a -26.9 (-36.7, -17.1) -72.3 (-83.9, -60.7) -33.3 (-40.4, -26.2) -45.5 (-53.5, -37.5)

Male (ref: Female)a 68.2 (62.5, 73.9) 67.9 (61.2, 74.6) 63.4 (58.9, 67.9) 64.4 (59.1, 69.7)

Age (ref: 30-34)a

- 35-39 17.5 (9.3, 25.7) 20.6 (11.0, 30.2) 9.4 (2.9, 15.9) 18.4 (11.0, 25.8)

- 40-44 35.5 (27.5, 43.5) 25.1 (15.7, 34.5) 25.6 (19.3, 31.9) 27.4 (20.0, 34.8)

- 45-49 33.5 (25.1, 41.9) 6.2 (-3.8, 16.2) 24.2 (17.5, 30.9) 16.6 (8.8, 24.4)

- 50-54 30.0 (21.6, 38.4) -9.5 (-19.5, 0.5) 22.2 (15.5, 28.9) 9.7 (1.7, 17.7)

Country of birth (ref: Sweden)a -78.5 (-86.1, -70.9) -72.4 (-81.4, -63.4) -20.1 (-26.4, -13.8) -19.9 (-27.3, -12.5)

Panel (ref: 2006)a

-2003 -5.3 (-12.7, 2.1) -9.5 (-18.3, -0.7) -6.9 (-12.8, -1.0) -9.4 (-16.3, -2.5)

-2004 -5.9 (-13.5, 1.7) -2.2 (-11.2, 6.8) -7.3 (-13.4, -1.2) -2.5 (-9.6, 4.6)

-2005 -7.9 (-15.5, -0.3) -6.8 (-15.8, 2.2) -6.0 (-12.1, 0.1) -3.3 (-10.4, 3.8)

Size of living region (ref: Larger cities)b

- Medium-sized municipalities -19.7 (-25.8, -13.6) -23.1 (-30.4, -15.8) -20.8 (-25.7, -15.9) -21.7 (-27.4, -16.0)

- Smaller municipalities -26.0 (-32.7, -19.3) -32.7 (-40.5, -24.9) -29.5 (-34.8, -24.2) -31.5 (-37.8, -25.2)

Education (ref: University/college)a

- High school -53.7 (-60.4, -47.0) -75.9 (-83.5, -68.3) - -

- Elementary school -110.3 (-119.9, -100.7) -139.9 (-151.3, -128.5) - -

- MS* High school -16.9 (-29.6, -4.2) -14.9 (-30.0, 0.2) - -

- MS* Elementary school -11.6 (-32.4, 9.2) -42.7 (-68.0, -17.4) - -

Type of Occupation (ref: Office work)b

- Managers - - 61.2 (48.3, 74.1) 95.7 (83.5, 107.9)

- Manual labour - - -64.5 (-69.6, -59.4) -71.2 (-77.1, -65.3)

- Unknown - - -138.4 (-149.6, -127.2) -165.2 (-183.4, -147.0)

- No Work - - -390.5 (-402.5, -378.5) -523.7 (-546.4, -501.0)

- MS*Managers - - 8.9 (-16.6, 34.4) -40.7 (-68.5, -12.9)

- MS* Manual labour - - -5.0 (-15.2, 5.2) -5.6 (-17.8, 6.6)

- MS* Unknown - - 8.9 (-12.7, 30.5) -24.7 (-60.2, 10.8)

- MS* No Work - - 27.0 (3.7, 50.3) 14.8 (-26.4, 56.0)

Sigma 131.1 (129.1, 133.1) 152.3 (149.9, 154.7) 100.6 (99.0, 102.2) 113.7 (111.9, 115.5)

N 10,137 9,928 10,138 9,928

N censored at zero 1,260 1,505 1,260 1,505

Bayesian information criterion (BIC) 114,739 111,979 107,898 104,333

a Time-invariant variables: measured at T0.

b Time-variant variables: measured at T0 or T+5.

Earnings: income from work and Student allowance, calculated in 2005 year's value. In 1,000 Swedish Crowns.

Year of inclusion; for people with MS: year of first MS diagnosis and for references: year of inclusion to one of four panels. Estimates from Tobit-models with lower bound set at zero (i.e., no earnings).

The covariates included in the regression models were (MS (yes/no), panel membership, sex, age at inclusion, country of birth, and type of living area), in addition to level of education and type of occupation. We did not use a statistical method to motivate the selection of these 'base' covariates, rather they were selected in light of the literature, for example, age at inclusion as a proxy for age at onset given that progression of MS is to some degree age-dependent (Confavreux & Vukusic, 2006 Brain). Age, alongside sex, was also used in selecting our matched reference group.

Regarding your comments about the possibility of GEE analyses, the aim of this study was to conduct an initial explorative test to investigate if there were any differences. The exact specifications of such potential differences are something that can be investigated further in other studies. The data allowed for a wide choice of statistical methods, and we agreed on Tobit regression models for this study. The main reason was that Tobit regression models are often considered in cases with clustering of values above and/or below certain thresholds. In cases without such clustering, results from a Tobit model are near identical to the corresponding results from an ordinary least squares regression (OLS). In this study, the proportion of clustered values (i.e., earnings=0) was about 12%. Further, given that all analysed covariates were categorical the results should be interpreted as a shift in level of mean earnings vs. the reference category. Since this is in line with the aim of the study we argue that this was the most appropriate method to explore the data.

However, we have crudely tried GEE models from year of diagnosis until end of study (T0-T+5). We included time as a categorical variable and an interaction time*MS, otherwise all other variables were the same as in the Tobit models, i.e. educational level and type of occupation in separate models. The GEE models were estimated using a one-step autoregressive correlation within the individual (AR(1)). This assumed that an individual's observations are less correlated the longer time has passed between the two observations. For example: For individual X, the earnings observed at T+2 are assumed to be more highly correlated with T+3 than T+4 or T+5.

Table 2 (Reviewer response): GEE regressions for earnings from year of diagnosis (T0) until end of study (T+5).

Model including Educational level Model including Type of Occupation

Estimate P-value Estimate P-value

MS -35.0 <.0001 -31.4 <.0001

Time (ref:T0) 0.0 0.0

T+1 7.9 <.0001 7.1 <.0001

T+2 15.8 <.0001 13.9 <.0001

T+3 23.3 <.0001 21.2 <.0001
T+4 31.0 <.0001 28.0 <.0001
T+5 35.9 <.0001 32.7 <.0001

MS*Time (ref:T0) 0.0 0.0

T+1 -16.5 <.0001 -13.1 <.0001
T+2 -20.2 <.0001 -14.1 <.0001
T+3 -25.5 <.0001 -18.7 <.0001
T+4 -32.6 <.0001 -23.3 <.0001
T+5 -35.4 <.0001 -25.0 <.0001

Overall, the estimates from the GEE models (presented here) would thus have resulted in the same interpretations of the differences between the two groups in T0 and T+5. Both regression techniques resulted in negative estimates which were in similar ranges at T0 and T+5. However, unlike the presented Tobit regression results in the manuscript, these results do not take into account the relatively high proportion of zeros (i.e., no earnings).

The GEE models we have presented here used time as a categorical variable, which is not ideal for describing the linearity. Our results give basis for suggesting that the specific nature of potential linearity could be further investigated in future studies which focus on longitudinal statistical methods, potentially by using piece-wise regressions with knots at 1 year before diagnosis and 1 year after.

We also agree that age is a very important variable in MS studies, given the age-dependency of MS prognosis (Confavreux C & Vukusic S. 2006 Brain). However, age has duly received much attention in the literature, and given our focus on education and occupation, we decided to match on age (and sex) when selecting the reference group, and age was accordingly included as a covariate in the regression models.

With regards to the comment concerning interactions between sex and education/occupation: We agree that such analysis would be of interest, especially given that the prevalence of MS is higher among women and sex differences in labour market attachment/earnings. However, to achieve this, either a complex three-way interaction is required in the Tobit regressions (MS*Sex*Education/occupation) or a stratified analysis including only the MS group. Such focus within the MS group is beyond the scope of this study. Future studies, stratifying by sex are needed, based on other aims than of this study. Hopefully, our results will inspire to conduct such studies, preferably based in gender theories.

minor points:

3. I 50: the estimate is SEK28,000 but the CI limits are negative, this is a bit awkward, I would just make them all positive and note that you did say "lower earnings" to give direction.

Authors' response: We have now substituted the confidence intervals in the abstract with p-values, to avoid confusion (See page 3, lines 52-53). In addition, we noticed a discrepancy in our Results section where we presented p-values in text and 95% confidence intervals in Table 4. We have accordingly amended the text in the Results section to handle this inconsistency (See page 20, lines 379-383).

4. I 54 alike should be like

Authors' response: Revised accordingly.

5. I 86 "may" I think you can cite literature to make a stronger claim that it does.

Authors' response: Thank you for this reflection. We have, now, added references to strengthen the sentence (Brundin et al, 2017 *Mult Scler*; Peason et al, 2017 *Acta Neurol Scand*; Krause et al, 2013 *Mult Scler*; Naci et al, 2010 *Pharmacoeconomics*; and Pflieger et al, 2010 *Mult Scler*), and we acknowledge that in most cases this claim is likely true. However, firstly, this sentence refers to the economic welfare of people with MS. Economic welfare is highly dependent on the, often public, social security system when one has work incapacity, and accordingly reduced earnings. In Sweden, studies have found that despite reductions in earnings (Wiberg et al, 2015 *Mult Scler*; and Landfeldt et al, 2018 *Value Health*) and increases in sickness absence and disability pension benefits (Wiberg et al, 2015 *Mult Scler*; and Gyllensten et al, 2016 *BMJ Open*) among people with MS, at least for the first years after diagnosis that overall economic welfare is largely unchanged due to the balancing of the sources of income (Landfeldt et al, 2018 *Value Health*; and Murley et al, 2018 *BMJ Open*). This also summarises the situation in Denmark (Pflieger et al, 2010 *Mult Scler*). Of course this can vary greatly from the situation in countries with other social security systems. However, to maintain our current focus of this paper on earnings from paid work, we agreed that adding such information so early in the paper could be misleading. Brundin et al, (2017 *Mult Scler*) present the findings for Sweden in a multi-country questionnaire, notably finding high numbers of part-time workers, and most (78%) of those in paid work self-reported that MS affects their productivity at work.

Secondly, there is large heterogeneity within the population with MS in any given setting, with regards to disease characteristics, for example, severity. Severity can be chronic, progressive but also with periods of small/undetected disability. Disability in terms of EDSS score has been found to be associated directly with earnings (Kavaliunas et al, 2015 *PLoS One*).

Thus we claim that may is the appropriate wording.

6. Table 4 N censored at zero reads 1,26

Authors' response: Thank you for bringing this typo to our attention. It has now been amended to the correct value of number censored (1,260).

Reviewer: 2

Reviewer Name: Claudia Pflieger

Institution and Country: Neurological Department, Aalborg University Hospital, Denmark Please state any competing interests or state 'None declared': None declared

Please leave your comments for the authors below Dear authors Thank you for this nicely written and systematically constructed paper on "earnings among people with multiple sclerosis compared to references, in total and by educational level and type of occupation: a population-based cohort study at different points in time"

Your findings are much in line with previous research and adds important knowledge to this interesting topic.

Authors' response: Thank you very much, we agree that this study provides much needed knowledge on the broader life situation of people with MS.

VERSION 2 – REVIEW

REVIEWER	john Pearson University Otago Christchurch New Zealand
REVIEW RETURNED	18-Feb-2019

GENERAL COMMENTS	A well powered investigation into the difference in income pre and post diagnosis of MS and with respect to a well chosen control group. Particularly interesting given the relatively high level of government income support in Sweden and high quality of the data available in Sweden. Models 1b and 2b which include the interaction with MS are important (and interesting!). Please include tables. I would also be interested to see a plot (box and whisker as in the included figures) of income by age stratified by MS (also by gender?). Income changes predictably over the life course and it is deviations from this pattern which indicate the impact of MS. Perhaps a Tobit model of this would be interesting too. I think it would be interesting to an international audience to have the age-income relationship explicitly described in Sweden, particularly the
---

	effect of MS on this relationship, thus addressing the issue of how MS affects income under the Swedish income/employment model. I like the inclusion of SA and DP. It would be interesting if the authors could tie this into the modelling, as it is its a bit of an awkward aside. A strength of this study is the number of males, this should allow comparisons of income by sex. In particular, if income x age x MS is plotted in this cohort are there differences for males and females and at what age or at what time since diagnosis? Statistics: Appropriate methods used, I like that the authors describe the population in absolute terms (effectively they have a census of PwMS for the criteria). Consider adding P values with the CIs in the text, I realise there are differing views on P values but let the readers make their own judgements. Income is often modelled on the log scale, was this considered in your data? A statement about this choice would aid completeness. In the text a statement is made about differences in variation over time, fig 2, this could be tested with a levene or KS type test. WE would typically expect that varaition would increaase with mean level of income, this is no the case in the ref group and it appears that there is a high varaition with the reduces incomes in the MS group. Is this effect explained by SA and DP, could this be explicitly modelled? Minor points: 1. Overall grammar is good, however there are some grammatical "glitches" which would benefit from the attention of a scientific writer. 2. p15 Make the subject of the paragraphs and sentences the thing you are describing, not the Table of Figure which is just a tool . Do this throughout the ms to enhance readability and interest to the audience. 3. The tables look good, try to remove jargon (eg T0 could be at inclusion, T5 could be 5 years, Beta is estimated difference.) 4. The plots look like they've been dumped out of SAS, fig 1 would probably be better with the histograms overlaid to allow comparisons across times. Describe the inferences you take from the plots better in the text.
--	---

VERSION 2 – AUTHOR RESPONSE

Reviewer(s)' Comments to Author:

Reviewer: 1

Reviewer Name: john Pearson

Institution and Country: University Otago Christchurch, New Zealand Please state any competing interests or state 'None declared': None Declared

Please leave your comments for the authors below

A well powered investigation into the difference in income pre and post diagnosis of MS and with respect to a well chosen control group. Particularly interesting given the relatively high level of government income support in Sweden and high quality of the data available in Sweden.

Models 1b and 2b which include the interaction with MS are important (and interesting!). Please include tables. I would also be interested to see a plot (box and whisker as in the included figures) of income by age stratified by MS (also by gender?). Income changes predictably over the life course and it is deviations from this pattern which indicate the impact of MS. Perhaps a Tobit model of this would be interesting too. I think it would be interesting to an international audience to have the age-income relationship explicitly described in Sweden, particularly the effect of MS on this relationship, thus addressing the issue of how MS affects income under the Swedish income/employment model.

I like the inclusion of SA and DP. It would be interesting if the authors could tie this into the modelling, as it is its a bit of an awkward aside.

A strength of this study is the number of males, this should allow comparisons of income by sex. In particular, if income x age x MS is plotted in this cohort are there differences for males and females and at what age or at what time since diagnosis?

Authors' Response: Thank you for the inspired and numerous constructive ideas for future studies of earnings over time among people with MS.

We are glad you found the results of the interactions MS*Education (Model 1b) and MS*Occupation (Model 2b) interesting, and have now included the table of the interactions as you requested. We have included the table as a supplement to the manuscript and modified the text regarding this table accordingly (See Supplement 1 and Results page 22).

The aim of this study was to, among people with MS and matched references, investigate differences in earnings over time and whether identified differences were associated with occupation and education, in addition to levels of part-time and full-time sickness absence (SA) and disability pension (DP). We are glad that our results inspire future research questions by the reviewer – which is of course what important studies of new areas do. However, we in this specific study with this aim cannot answer all study questions. Our current aim was already three-fold and with two study outcomes (pre-tax earnings and proportions on SA/ DP). We are in full agreement with the reviewer that based on this study, future studies, stratified by sex and investigating the age-earnings relationship explicitly are needed, with specific aims concerning this. We have now expressed our suggestions for future studies looking into these questions in our revised discussion (See page 26 lines 790-797).

It is always prudent and interesting to check the data in many ways before performing the main analysis. There are of course many factors that are associated with earnings. We have focused our analysis in depth and discussed two factors, rather than breadth, to include a broader range of possible factors behind the heterogeneity of earnings among people recently diagnosed with MS. Our focus was specifically on educational level and occupation, two modifiable factors that could help inform the knowledge base for strategies for these individuals diagnosed with MS. Therefore, we have not included other plots (or series of) of earnings among people with MS over time by age and sex. However, we have to some extent previously showed earnings by age and gender in the Swedish

MS-population namely here: Murley et al, Disposable income trajectories of working aged individuals with diagnosed multiple sclerosis, *Acta Neurol Scand* 2018;138. 490-499). We have now added this as a reference in the discussion (See page 26, lines 793-795). Similarly, regarding the comment about interactions between sex and earnings: We agree that such analysis stratified by sex would be of interest, especially given that the prevalence of MS is higher among women than men and that there are known sex differences in labour market attachment/earnings. However, to achieve this, either a complex three-way interaction is required in the Tobit regressions (MS*Sex*Education/occupation) or a stratified analysis including only the MS group. Such focus within the MS group is beyond the scope of this study. Future studies, stratifying by sex are needed, based on other aims than of this study. We have, however, attempted it with a different approach in the previous publication (Murley et al 2018). We have added comments about this in the discussion of future studies. (See page 26, lines 793-795).

Given that in addition to earnings, SA and DP benefits are two very important sources of the total incomes of someone with a disabling chronic disease in Sweden may have, at least in a long-term perspective, they were also included as an outcome in this study. Of course there are many ways SA and DP could be explored further in future studies. This is discussed in the Discussion section (see pages 24-25, lines 741-772).

Statistics:

1) Appropriate methods used, I like that the authors describe the population in absolute terms (effectively they have a census of PwMS for the criteria). Consider adding P values with the CIs in the text, I realise there are differing views on P values but let the readers make their own judgements.

Authors' Response :Thank you. There are several different methods of presenting statistically significant/insignificant results. We have now included asterisks in Table 4, in addition to the 95% confidence intervals, to indicate where the estimated differences in earnings were significantly different from zero ($p < 0.05$). This is consistent with how results are presented in Table 2. We prefer to keep the CIs throughout the Results section text as they are more informative than P-values.

2) Income is often modelled on the log scale, was this considered in your data? A statement about this choice would aid completeness.

Authors' Response: Thank you for this comment. We have now added a description of our decision not to transform the outcome variable in the Methods (page 8 and lines 261-263).

It is always a decision whether one should or should not transform the outcome variable (here earnings). We decided to measure earnings as an untransformed continuous variable. Using a log-scale for earnings would have resulted in an interpretation of the percentage difference between the groups. While this may have different statistical properties and implications for analysis, it was not used here and our statistical decisions, presentation and interpretation of results were based on actual differences in earnings between the groups.

3) In the text a statement is made about differences in variation over time, fig 2, this could be tested with a Levene or KS type test. We would typically expect that variation would increase with mean level of income, this is not the case in the ref group and it appears that there is a high variation with the reduced incomes in the MS group. Is this effect explained by SA and DP, could this be explicitly modelled?

Authors' Response: Thank you for this comment. The variation observed very likely does depend on SA and DP, but there are also likely very many other factors which also contribute to the variation. For example, clinical factors, given the wide heterogeneity in the disease progression. We have elaborated on factors behind this heterogeneity in the discussion (See pages 24-25, lines 741-772). However, to go further and model this is beyond this study's aims.

Minor points:

9) 1. Overall grammar is good, however there are some grammatical "glitches" which would benefit from the attention of a scientific writer.

Authors' Response: We have taken this opportunity to edit the manuscript and have made minor changes throughout. For example, we have revised the outcome variables section in the Methods (See pages 8-9, lines 255-295) to clarify the conceptual and operational definitions of the outcome variables for this study. Namely, earnings, which we study as the pre-tax earnings in order to reduce possible confusion with income (which includes after tax sums of earnings, transfer payments and capital incomes).

10) 2. p15 Make the subject of the paragraphs and sentences the thing you are describing, not the Table of Figure which is just a tool. Do this throughout the ms to enhance readability and interest to the audience.

Authors' Response: Thank you. We have now revised the Results section with topic sentences to improve the narrative of the results section in light of your comment (See Results section pages 13-22).

11) 3. The tables look good, try to remove jargon (eg T0 could be at inclusion, T5 could be 5 years, Beta is estimated difference.)

Authors' Response: Thank you. We have accordingly revised the tables to guide the reader to interpret the results as per your suggestion (See Results section, Table 4 and new Supplementary Table 1). Furthermore, we have made minor revisions to language throughout the manuscript to improve readability.

12) 4. The plots look like they've been dumped out of SAS, fig 1 would probably be better with the histograms overlaid to allow comparisons across times. Describe the inferences you take from the plots better in the text.

Authors' Response: Thank you for the suggestion. We have now revised both figures to reduce jargon and aid readability (See Figures 1 and 2).

However, following your suggestion regarding overlaying the two figures would lead to a very complicated figure which may reduce readability. There is a lot of information which could be obscured. Further, Figure 1 is not stratified by MS, and therefore this limits how much we should interpret. Rather, the inclusion of Figure 1 was to describe and justify the use of the Tobit models, i.e., to guide statistical choices than as a result of itself. We have added a description of Figure 1 regarding the zero clustering of earnings that guided our choice of methods (See page 12, lines 423-424).

VERSION 3 - REVIEW

REVIEWER	John Pearson University Otago Christchurch New Zealand
REVIEW RETURNED	07-Apr-2019

GENERAL COMMENTS	Very nice paper, thanks for addressing my concerns. Best wishes.
--